# Combining antimiR-25 and cGAMP Nanocomplexes Enhances Immune Responses via M2 Macrophage Reprogramming

**DOI:** 10.3390/ijms252312787

**Published:** 2024-11-28

**Authors:** Marija Petrovic, Oliwia B. Majchrzak, Rihana Amreen Mohamed Hachime Marecar, Annick C. Laingoniaina, Paul R. Walker, Gerrit Borchard, Olivier Jordan, Stoyan Tankov

**Affiliations:** 1Institute of Pharmaceutical Sciences of Western Switzerland (ISPSO), Faculty of Science, University of Geneva, 1206 Geneva, Switzerland; oliwia.majchrzak@unige.ch (O.B.M.); gerrit.borchard@unige.ch (G.B.); olivier.jordan@unige.ch (O.J.); 2Translational Research Center in Onco-Hematology (CRTOH), Faculty of Medicine, University of Geneva, 1206 Geneva, Switzerland; paul.walker@unige.ch

**Keywords:** cGAMP, antimiR-25, antagomir-25, STING pathway, nanomedicine, cancer immunotherapy, polymeric nanoparticles, PAMAM, extracellular vesicles, EVs

## Abstract

Glioblastoma (GBM) is an aggressive brain cancer with a highly immunosuppressive tumor microenvironment (TME), invariably infiltrated by tumor-associated macrophages (TAMs). These TAMs resemble M2 macrophages, which promote tumor growth and suppress immune responses. GBM cells secrete extracellular vesicles (EVs) containing microRNA-25, which inhibits the cGAS-STING pathway and prevents TAMs from adopting a pro-inflammatory M1 phenotype. This study characterizes antimiR-25/cGAMP nanocomplexes (NCs) for potential therapeutic applications. A particle size analysis revealed a significant reduction upon complexation with antimiR-25, resulting in smaller, more stable nanoparticles. Stability tests across pH levels (4–6) and temperatures (25–37 °C) demonstrated their resilience in various biological environments. Biological assays showed that antimiR-25 NCs interacted strongly with transferrin (Tf), suggesting potential for blood–brain barrier passage. The use of cGAMP NCs activated the cGAS-STING pathway in macrophages, leading to increased type I IFN (IFN-β) production and promoting a shift from the M2 to M1 phenotype. The combined use of cGAMP and antimiR-25 NCs also increased the expression of markers involved in M1 polarization. These findings offer insights into optimizing antimiR-25/cGAMP NCs for enhancing immune responses in GBM.

## 1. Introduction

Glioblastoma (GBM) is one of the most aggressive primary brain cancers [1]. Rapidly growing cancer cells consume high amounts of oxygen supplied through tumoral blood vessels, leading to the creation of hypoxic conditions in the GBM tumor microenvironment (TME) [2]. As an immune-privileged organ, the brain expresses a tightly regulated immune response, which renders its environment particularly immunosuppressive. Characteristically for cold tumors, immune cells poorly infiltrate the GBM’s TME [3].

There is significant diversity both within individual tumors and among different GBM tumors. However, one common feature in GBM is the infiltration of myeloid cells (macrophages and microglia), the majority of which are tumor-associated macrophages (TAMs). TAMs have drawn a great deal of scientific attention due to their phenotypical resemblance to the immune-suppressive, alternatively activated M2 macrophages [4,5], which can antagonize T-cell activation and allow the tumor to evade the immune response. Moreover, hypoxia in the TME promotes immunosuppression, which further favors tumor development and growth [5].

Hypoxia in the TME leads to the upregulation of specific microRNAs (miRs), small non-coding RNA sequences that play a critical role in post-transcriptional gene regulation and indirectly shape the TME [6]. These upregulated miRs, including hypoxia-induced miR-25, can be transported into the extracellular matrix via extracellular vesicles (EVs), where they are subsequently taken up by TAMs. For instance, miR-25, overexpressed under hypoxic conditions in cancer cells, is packaged into EVs and delivered to macrophages. Once inside macrophages, miR-25 inhibits the cGAS-STING (Cyclic GMP-AMP synthase–STimulator of INterferon Genes) pathway—a key pathway that promotes the transition of macrophages from an immunosuppressive M2-like phenotype to a pro-inflammatory M1 phenotype needed for effective tumor cell eradication [1,7]. Additionally, EVs may carry other molecules, including ligands and receptors, that facilitate direct interactions with macrophage receptors, such as those on bone marrow-derived macrophages (BMDMs), influencing their behavior and contributing to the immunosuppressive effects seen in the TME. In GBM, for example, EVs were shown to carry PD-L1, which can engage PD-1 receptors on macrophages, suppressing macrophage pro-inflammatory activity and fostering a tolerogenic, immunosuppressive phenotype that supports tumor growth [8,9]. In contrast, some EVs contain pro-inflammatory ligands, such as TNF-α or HMGB1, which bind to TNFR or TLRs on macrophages, initiating an inflammatory response that promotes macrophage cytotoxicity and an anti-tumor immune response [10]. This balance between activating and inhibitory signals, mediated through EV ligand–receptor interactions, shapes macrophage polarization and affects the immune landscape within the TME, influencing cancer progression and responses to therapy.

The cGAS-STING pathway activation in antigen-presenting cells (APCs) is an important mechanism of innate immune activation against the tumor, due to the activation of a cascade reaction through TBK1 and IRF3 phosphorylation, consequent transcription, and type I interferon production. The cGAS-STING pathway is activated by its canonical ligand, 2′3′-cyclic guanosine monophosphate–adenosine monophosphate (cGAMP), or by double-stranded DNA (dsDNA) present in the cytosol [11]. Chromosomal and genomic instability in cancer cells results in high amounts of cytosolic dsDNA, which can potentially activate the STING pathway in cancer cells or in other cells if exported. However, the miRNAs co-exported by cancer cells can impede this activation process [7]. As a result, M2-like TAMs may not efficiently repolarize towards an M1 phenotype and participate in anti-tumor responses.

The standard of care of GBM treatment includes maximal surgical resection followed by adjuvant radiotherapy and chemotherapy [12]. The success rate of surgical interventions is case-dependent: location, size, and sensitive brain regions surrounding the resection area must be considered. It is important to mention that tumor hypoxia significantly contributes to increased radio- and chemotherapy resistance [1]. To address these challenges, novel therapeutic approaches must be investigated to treat GBM, which is why immunotherapy-based approaches are currently extensively explored.

Nevertheless, the use of nucleotides such as antimiR-25 or cGAMP remains challenging, mainly due to their instability influenced by enzymatic degradation, or because of the complexity related to cell transfection. The latter is partially due to the repulsions between the negative charges of the nucleic acids and the cell membrane.

Numerous nanosystems are currently being studied for GBM treatment, namely, polymeric nanoparticles, polymeric micelles, dendrimers, liposomes, and solid lipid nanoparticles [13]. Poly(amidoamine) (PAMAM) dendrimers offer several benefits, including a positive charge at low pH levels (corresponding to TME conditions) for effective oligonucleotide complexation, customizable surface groups facilitating targeting and low immunogenicity. Moreover, tertiary amines in the backbone potentially aid nanoparticle escape from endosomal compartments, possibly through a proton sponge effect [14]. PAMAMs are classified based on their degree of branching into generations. Higher generations exhibit an augmented number of terminal primary amine groups, leading to enhanced transfection efficiency. This improvement stems from both increased global positive charges, fostering interactions with the negatively charged cell membrane, and a greater abundance of terminal groups available for conjugation with desired targeting moieties. However, a drawback of higher-generation PAMAM is the associated rise in cytotoxicity [14]. Hence, a preference is given to lower generations of PAMAMs, with the prioritization of generation 3 PAMAMs based on our prior investigation [15]. Our previous study demonstrated that PAMAM generation 3 exhibited lower toxicity and higher transfection efficiency in comparison to PAMAM generation 4 [16,17]. Moreover, we implemented the partial neutralization of terminal primary amines to effectively reduce their toxicity by conjugating the amines with ligands targeting a selected receptor expressed by M2 macrophages. To this end, we designed PAMAM dendrimers grafted with D-glucuronic acid as carriers for oligonucleotides [15]. One of the differences between M2 and M1 macrophages is the presence of cell surface receptors such as the C-lectin group of receptors, specifically the mannose receptor (CD206) [5,18], which is commonly activated by selected monosaccharides such as mannose, fucose, and glucuronic acid produced by various microorganisms such as yeasts, bacteria, and viruses [19]. In our attempt to replicate a physiological microorganism-like signal and attract APCs for compound uptake and processing, we conjugated glucuronic acid to PAMAM. This conjugation led to the development of synthetic nanocarriers called PG3, exhibiting an affinity towards CD206 and demonstrating low toxicity [20].

The main goal of this study was to create M2-targeting nanocomplexes (NCs) based on a modified PAMAM scaffold capable of carrying antimiR-25 as a novel immunotherapeutic modality, meeting critical formulation parameters on one hand, while triggering the cGAS-STING pathway and stimulating a macrophage shift from the M2 to M1 phenotype on the other hand.

## 2. Results and Discussion

### 2.1. Charcterization of antimiR-25 Nanocomplexes (NCs)

The fundamental characterization of nanoparticles involves the measurement of size and zeta potential. Additional imaging helps to determine the morphology of nanoparticles. Notably, we maintained a fixed N/P ratio of 2/1 and 1/1, representing the molar-charge ratio between the amine groups of the polymer and the phosphate groups of either antimiR-25 or cGAMP, respectively (Figure 1). An N/P ratio of 1/1 has a net neutral charge based on its equal number of phosphate groups and amine groups and is considered a control for the study. Given that the cell membrane is negatively charged, the 2/1 ratio is deemed to better transfect cells due to its net positive charge and this ratio is also the minimal requirement for cellular internalization [21]. In a clinical setting, the preferred administration route of the antimiR-25 NCs would be intravenous injection, assuming that the NCs had the capability of traversing the blood–brain barrier (BBB).

#### 2.1.1. NC Size Measurements

Regarding DLS measurement (Figure 2a), the PG3 control contains significantly larger particles than after complexation with antimiR-25 at both ratios. Mean sizes of PG3 NCs of 2/1 and 1/1 ratios were 222 nm and 256 nm, respectively. Complexation leads to a decrease in the measured nanoparticle size, probably due to the polymer’s greater dispersibility and therefore the presence of fewer aggregates in the system. As for NTA values, a statistically significant difference is absent before and after complexation for NP ratios. This can be a consequence of the intensity-average detection performed by DLS, focusing on larger particles such as aggregates. However, NTA size distribution graphs (Appendix A) show that PG3 alone presents multiple populations of different sizes. While most particles range from 80 to 200 nm in diameter, smaller populations of larger particles in the range of 400 to 800 nm are also found. However, upon complexation with antimiR-25, populations of larger particles are no longer detectable, and the distribution approaches a monomodal form. This observation confirms the DLS size analysis where less aggregation occurs after complexation. Moreover, there is a significant difference in size between DLS and NTA measurements for PG3 alone, while this is not the case for the NCs (*p* < 0.05). These results indicate that the PG3 is stabilized when complexed with antimiR-25, further supporting successful dendrimer–antimiR-25 complexation.

#### 2.1.2. NC Zeta Potential

Figure 2b shows significantly greater zeta potential in the control sample (PG3 CTRL) compared to antimiR-25 NCs at both N/P ratios, which is an expected phenomenon as upon PG3 complexation with antimiR-25, a positive net surface charge of PG3 should be compensated for due to the intrinsic antimiR-25’s negative charge. Interestingly, both 2/1 and 1/1 antimiR-NCs show a negative ZP rather than the theoretically expected positive and neutral values (Figure 2b). A possible explanation can be deduced from the organization of the particles, positive charges being constrained to the PG3 surface surrounded by negatively charged antimiR-25 molecules. As demonstrated in our prior research, the specific zeta potential values derived from both DLS and NTA show a correlated trend under different conditions but are visibly distinct from each other [22].

#### 2.1.3. NC Number of Particles

An analysis of the number of particles aids in characterizing the degree of the aggregation of PG3 before complexation with antimiR-25. Using NTA, we observed (Figure 2c) a statistically significant increase in the number of particles upon complexation at a ratio of 2/1 compared to the controls, antimiR-25 and PG3. Dendrimers alone aggregate, reducing the number of particles detected by NTA, whereas upon complexation with antimiR-25, the size of the aggregates is reduced. This correlates with the DLS size analysis: preparations at a 2/1 ratio tend to have more particles than for the 1/1 ratio, probably because there is more dispersion at 2/1 than 1/1, and a lower number of polymer aggregates. Complexation under ideal conditions should result in a lower particle count in comparison to the starting number.

#### 2.1.4. Shape of NC

Finally, SEM results support findings on the size and number of particles. As seen in Figure 2d, PG3 alone is prone to form crystal-like structures. However, once antimiR-25 is complexed to PG3, NC images, at ratios 2/1 and 1/1, respectively, present smaller spherical particles. These results correlate with DLS and NTA characterization and might support complexation.

In conclusion, antimiR-25 NCs were characterized in terms of nanocomplexation; based on ZP and size characterization, both the ZP and size of NCs significantly decrease compared to control PG3. Mean sizes of PG3 NCs at 2/1 and 1/1 ratios were 222 nm and 256 nm, respectively, which fall in the size range of 200 to 300 nm, compatible with efficient uptake by macrophages [23]. Furthermore, NCs at an N/P ratio of 2/1 seem less prone to aggregation due to lower PDI values and a higher number of particles compared to the 1/1 ratio, as confirmed by the above-mentioned orthogonal techniques.

#### 2.1.5. pH Stability

Next, to explore critical formulation parameters, we investigated stability of NCs over time, at different pH and temperature values, as well as the encapsulation efficiency (EE) of the antimiR-25 at pH values corresponding to the selected intracellular compartments (lysosomes at pH 4 and endosomes at pH 5) important in the cell internalization process. Incubating nanoparticles at temperatures ranging from 25 to 37 °C was chosen to predict NC biostability and access their ability for endosomal escape.

Particle size did not significantly vary between different pH values (Figure 3a). In addition, as presented in Figure 3b, the pH did not affect encapsulation efficiency in the pH range between 4 and 6. We conclude that NCs are stable in this pH range. Consequently, endosomal escape would be expected to occur through a mechanism other than that of the proton sponge effect. An example of such a mechanism may be membrane destabilization involving the interaction of cationic polymer amines with negative charges of endosomal phospholipids, inducing membrane rupture and NC release into the cytosol [24].

#### 2.1.6. Temperature Stability

A gradual increase in temperature at a rate of 0.5 °C/min from 25 °C to 37 °C (Figure 3c) did not affect the size of antimiR-25 NCs. Based on this, we demonstrate that the stability of the NC particles, after administration at room temperature, should not change once the particles reach human body temperature.

#### 2.1.7. Storage Stability

The purpose of this experiment was to identify the NCs with the best stability over time and to provide perspectives on stability optimization during storage. AntimiR-25 NCs were stored under different conditions and analyzed after 1 week, after 1 month, and after freeze drying. The size, ZP, and number of particles were measured by DLS and NTA (Figure 4). Freeze drying was the best method to retain stable NC parameters, leading to similar sizes compared to freshly prepared NCs (Figure 4b). However, in all the cases, ZP significantly differed from freshly prepared NCs (Figure 4a). Only lyophilized NCs showed no significant differences in terms of size between NTA and DLS (Figure 3b), identifying lyophilization as the optimal technique for storage stability.

Overall, the results on the stability of antimiR-25 NCs (PG3-antimiR-25 at N/P 2/1 ratio) provide insights into product optimization, particularly the influence of pH and temperature. Since size and EE are not influenced by pH variation from 4 to 6, PG3 2/1 NCs were relatively stable in the selected pH range of interest. Regarding temperature stability, size did not increase significantly when temperature gradually increased from 25 °C to 37 °C. To summarize, antimiR-25 NCs were quite resistant to different pH values (4–6) and temperature (25–37 °C). NCs subjected to freeze drying were comparable to the freshly prepared particles. Nevertheless, stability upon storage should be further investigated with the use of different cryoprotectants or by using lower freezing rates [25,26] to yield better results.

### 2.2. The antimiR-25 NC Interactions with Transferrin

To predict the targeting and penetration of the BBB, we explored the interactions of antimiR-25 NCs with a selected serum protein, transferrin (Tf). In the prospect of targeting GBM, we speculate that transferrin receptor-mediated transcytosis (RMT) could facilitate BBB passage. Human holo-transferrin (HTF) affinity was measured by a spectroscopic analysis of the HTF binding isotherm with antimiR-25-loaded PAMAM. The measured Ka between antimiR-25 NCs (N/P ratio 2/1) and HTF was 2.54 × 10^7^ M^−1^ (Figure 5c), which was greater than the typical binding constants found in the literature for fourth-generation unmodified PAMAM binding with proteins such as albumin (1.4 × 10^5^ M^−1^) [27,28,29]. PG3 complexation with antimiR-25 on top of PAMAM’s glucoronation might be the explanation for the observed Ka values. To confirm those findings, Videodrop NC size examination was performed as a live-time size-tracking technique. Nanocomplexes alone exhibited a relatively monodispersed size distribution with a peak around 100 nm (Figure 5a), indicating a uniform particle nature. In contrast, mixed antimiR-25 NC-HTF samples (Figure 5b) showed a broader size distribution with a peak around 300 nm, suggesting increased inter-particle interactions. More dispersed size distribution implies that HFT might coat antimiR-25 NCs to varying extents in accordance with the tracked particle sizes.

### 2.3. cGAS-STING Pathway Activation

To directly trigger the cGAS-STING pathway and stimulate a macrophage polarization shift from the M2 to M1 phenotype, we physically combined antimiR-25 NCs with cGAMP NCs to study their joint properties in vitro on murine BMDMs exposed to hypoxic GBM-derived EVs and incubated at different oxygen levels, mimicking the GBM To assess the effect of antimiR-25 (a negative regulator of miR-25) NCs and cGAMP (a STING pathway activator) NCs on cGAS-STING pathway activation and macrophage polarization status, we conducted the following analyses:

(i) Measurement of IFN-β levels in macrophages treated with increasing concentrations of cGAMP and antimiR-25 NCs (Figure 6);

(ii) Analysis of the expression levels of selected macrophage signature genes in macrophages treated with single or combined NCs under hypoxic (1% O_2_) or normoxic (21% O_2_) conditions (Figure 7 and Figure 8, respectively); and

(iii) Measurement of IFN-β levels in M0 and M2 polarized macrophages treated with cGAMP NCs and antimiR-25 NCs to investigate potential synergistic effects between cGAMP and antimiR-25 (Figure 9).

We used macrophages polarized toward M0 or M2, treated for 24 h under 1%, 5%, or 21% O_2_. These oxygen concentrations were selected to mimic hypoxic tumor zones (1% O_2_), physioxic brain conditions (5% O_2_), and standard laboratory conditions (21% O_2_). This short-term deprivation of O_2_ is sufficient to induce most HIF-dependent adaptations in macrophage function, but it is possible that longer-term hypoxia in vivo would amplify the changes.

#### 2.3.1. IFN-β Secretion Levels from Unpolarized M0 and M2-Polarized BMDMs Treated with antimiR-25 NCs and cGAMP NCs

We previously demonstrated the major impact of cGAMP NCs (cGAMP complexed with PG3) in STING pathway activation, which was indirectly monitored via IFN-β quantification [7]. Here, we measured the IFN-β secretion levels from unpolarized M0 and M2-polarized BMDMs (Figure 6a and Figure 6b, respectively) treated with antimiR-25 NCs (0, 0.072, 0.143, 0.215, or 0.286 µg/mL) and cGAMP NCs (0, 25, 50, 75, or 100 µg/mL). The BMDMs showed a similar trend in response to increasing concentrations of cGAMP NCs, regardless of their phenotype. All cGAMP NCs demonstrated significant capacity to stimulate IFN-β production at a concentration of 25 µg/mL. However, antimiR-25 NCs did not significantly contribute to IFN-β expression. The effect observed was dose-dependent, revealing a pattern where increasing cGAMP concentrations correlated with higher IFN-β production, plateauing at 75 µg/mL.

**Figure 6 ijms-25-12787-f006:**
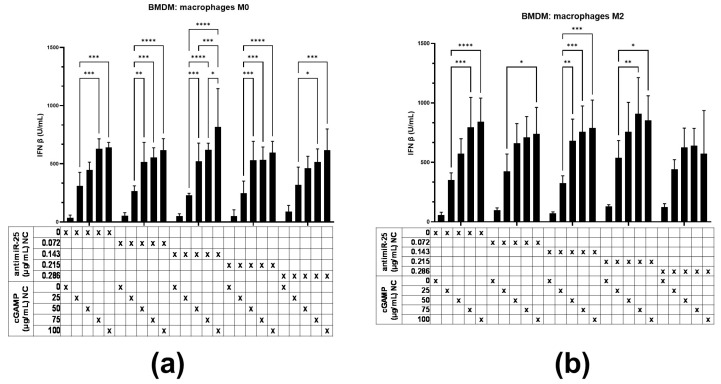
IFN-β secretion levels in BMDMs upon exposure to increasing cGAMP and antimiR-25 NCs. Unpolarized M0 (**a**) or polarized towards M2 (**b**) BMDMs treated with antimiR-25 (0, 0.072, 0.143, 0.215, or 0.286 µg/mL) and cGAMP (0, 25, 50, 75, or 100 µg/mL) NCs. x-denotes the combination of formulation parameters being investigated. Data are presented as the mean ± SD of three independent experiments and comparisons were made using an unpaired *t* test. * *p* < 0.05, ** *p* < 0.005, *** *p* < 0.001, and **** *p* < 0.0001.

#### 2.3.2. Expression of Genes Associated with M2/M1 Macrophage Polarization

Additionally, we measured the expression of genes associated with M2/M1 macrophage polarization (*Arg1* and *Mrc1* corresponding to the M2-associated macrophage gene signature, and *Cxcl10*, *Ifna*, *Il1b*, and *Nos2* indicating the M1-like macrophage gene signature) in M2-polarized and M0-unpolarized BMDMs. The macrophages were treated with antimiR-25 NCs, cGAMP NCs, or a combination of both, in the presence of hypoxic GBM-derived EVs under 1% O_2_ (Figure 7) and 21% O_2_ (Figure 8) conditions. At 1% O_2_, M0 macrophages polarized towards an M1-like phenotype in response to exposure of both cGAMP and antimiR-25 NCs, either individually or together, as shown by significantly decreased *Mrc1* expression and increased *Cxcl10* and *Ifna* expression. Conversely, shifting an already defined M2 phenotype towards an M1-like one was more challenging, as the M2-like signature was not strongly affected by the incubation conditions. However, M1-associated genes tended to increase after incubating M2 macrophages with NCs. Notably, the combination of antimiR-25 and cGAMP NCs significantly upregulated *Cxcl10* and antimiR-25-alone treatment upregulated the *Ifna* gene expression (Figure 7).

**Figure 7 ijms-25-12787-f007:**
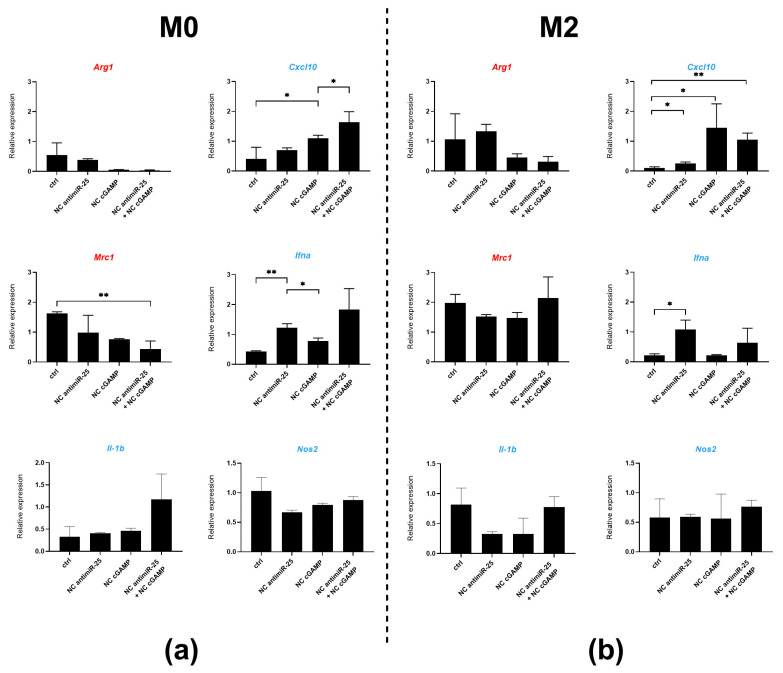
The antimiR-25 and cGAMP NC treatments reprogram gene expression in BMDMs under hypoxic conditions. The expression of M2-associated markers (*Arg1* and *Mrc1*) (in red) and M1-associated markers (*Cxcl10*, *Il1b*, *Nos2*, and *Ifna*) (in blue) measured by RT-qPCR. The left panel represents the relative gene expression in unpolarized M0 BMDMs (**a**) and the right panel represents M2-polarized BMDMs (**b**). Macrophages were treated with antimiR-25-coupled NCs (0.143 µg/mL) and/or cGAMP-coupled NCs (100 µg/mL) or empty NCs (ctrl) for 24 h in the presence of hypoxic GBM-derived EVs at a 1% O_2_ level. Data are presented as the mean ± SD of three independent experiments and comparisons were made using an unpaired t test. * *p* < 0.05 and ** *p* < 0.005.

At 21% O_2_, macrophages responded less to the NC treatment, with a modest increase in the expression of the M1-associated gene *Cxcl10* and a decrease in the expression of the M2-associated gene *Mrc1* in M0 macrophages. However, we observed a significant increase in *Nos2* expression after treatment with cGAMP NCs (in both M0 and M2 macrophages) and with the combination of antimiR-25 and cGAMP NCs (in M0 macrophages). Similarly to the result at 1% O_2_, M2-polarized macrophages under 21% O_2_ were not strongly impacted by the NC treatment, with only a modest trend for *Cxcl10* upregulation by cGAMP NCs, and *Nos2* upregulation by antimiR-25 NCs, with no synergistic effect (Figure 8).

**Figure 8 ijms-25-12787-f008:**
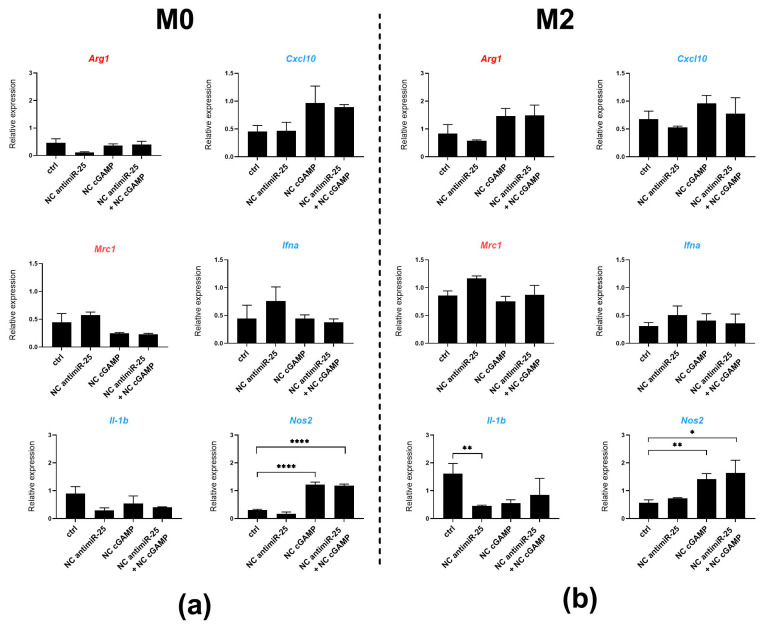
The antimiR-25 and cGAMP NC treatments modulate *Nos2* gene expression in BMDMs under normoxic conditions. The expression of M2-associated markers (*Arg1* and *Mrc1*) (in red) and M1-associated markers (*Cxcl10*, *Il1b*, *Nos2*, and *Ifna*) (in blue) measured by RT-qPCR. The left panel represents the relative gene expression in unpolarized M0 BMDMs (**a**) and the right panel represents M2-polarized BMDMs (**b**). Macrophages were treated with antimiR-25-coupled NCs (0.143 µg/mL) and/or cGAMP-coupled NCs (100 µg/mL) or empty NCs (ctrl) in the presence of hypoxic GBM-derived EVs at a 21% O_2_ level. Data are presented as the mean ± SD of three independent experiments and comparisons were made using an unpaired t test. * *p* < 0.05 and ** *p* < 0.005 and **** *p* < 0.0001.

#### 2.3.3. The Impact of Oxygen Levels on M0 and M2 Macrophages

By varying oxygen levels, we recapitulated different tumor TME-like scenarios, with 1% O_2_ levels to mimic hypoxic conditions in the GBM TME, whereas 21% O_2_ levels represent atmospheric conditions commonly used for in vitro studies.

To investigate how macrophages are affected by NCs under normoxia or hypoxia (as in the TME), we used in vitro unpolarized M0 or M2-polarized BMDMs. These cells were treated with empty PG3 NCs, antimiR-25 NCs, cGAMP NCs, or a combination of antimiR-25 and cGAMP NCs (Figure 9). The results showed that M0 macrophages were particularly sensitive to cGAMP NCs, secreting significantly greater amounts of IFN-β under both 1% and 21% O_2_. M2 macrophages secreted lower levels of IFN-β, with slightly higher levels secreted under 21% O_2_ conditions. Notably, M0 macrophages secreted over five times more IFN-β than M2 macrophages following cGAMP NC treatment, regardless of atmospheric or hypoxic O_2_ conditions. Free (non-encapsulated) cGAMP was also sufficient to stimulate IFN-β production (Appendix A); however, embedding it in PG3 NCs reduces its toxicity and would protect it from enzymatic degradation in the circulatory system. Thus, PG3 NCs served a dual purpose: reducing toxicity and enhancing the stability of the nanocarriers. In summary, although antimiR-25 did not significantly promote IFN-β secretion, cGAMP NC treatment robustly increased IFN-β production in all O_2_ conditions, especially by M0 macrophages.

**Figure 9 ijms-25-12787-f009:**
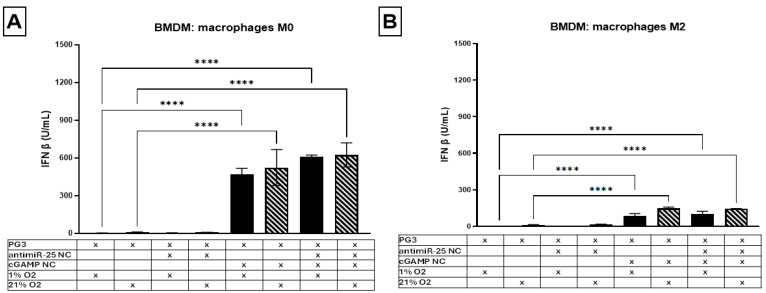
cGAMP NCs induce IFN-β secretion in M0 and M2 macrophages under 1% and 21% O_2_. The quantification of IFN-β secretion by BMDMs polarized to M0 (**A**) and M2 (**B**) upon incubation at different oxygen levels (1% O_2_ or 21% O_2_). The BMDMs were treated with empty NCs (PG3) or NCs loaded with antimiR-25 and/or cGAMP, x-denotes the combination of formulation parameters being investigated. Data are presented as the mean ± SD of three independent experiments and comparisons were made using 2-way ANOVA. **** *p* < 0.0001.

Overall, NCs coupled with cGAMP and antimiR-25 activated the cGAS-STING pathway in macrophages in vitro, leading to enhanced IFN-β secretion and a shift in gene expression towards an M1-like phenotype. cGAMP played the dominant role in this process, though the addition of antimiR-25 NCs elevated the expression of some M1-associated genes. This finding underscores the critical role of cGAS in cGAS-STING-induced macrophage polarization.

## 3. Materials and Methods

### 3.1. Chemicals

PAMAM with an ethylenediamine core of the third generation (Mw = 6937 g/mol, 20 wt% in methanol) was purchased from Sigma-Aldrich, St. Louis, MO, USA; D-glucuronic acid (D-Glu) from Alfa Aesar (Karlsruhe, Germany), and 2′3′-cGAMP, was obtained from Invivogen (Toulouse, France). The antimiR-25 was purchased from ThermoFisher (Waltham, MA, USA), ID MC10584; the miR-25 inhibitor sequence is UCAGACCGAGACAAGUGCAAUG.

### 3.2. Nanocarrier Synthesis

D-glucuronic acid was covalently conjugated to PAMAM following an EDC-NHS-guided amidation strategy taking place between PAMAM’s terminal primary amines and the carboxyl groups present in D-glucuronic acid, which resulted in PG3 as described and characterized in our previous work [15].

### 3.3. Nanocomplexation

Nanocomplexes were formed through ionic gelation in which the antimiR-25 in a solution was added onto the PG3 in a solution according to the selected N/P ratio and incubated while vortexing for 2 h at RT, adapted from [30]. PG3 alone (PG3 CTRL) as well as antimiR-25 alone were prepared as controls. The antimiR-25 concentration was fixed at certain values and the corresponding concentrations of PG3 were calculated depending on the desired N/P ratio. Therefore, for 143 ng/mL of antimiR-25, a solution of 200 ng/mL of PG3 was prepared in case of the 2/1 N/P ratio and twice less of PG3 for the 1/1 ratio. cGAMP NCs were prepared at an N/P ratio of 2/1 by mixing 0.1 mg/mL cGAMP and 250 µg of PG3.

### 3.4. Systematic Nanocharacterization

Several critical quality attributes (CQAs) of the particles were investigated in order to characterize produced nanocomplexes, including net surface charge as a function of zeta potential (ZP), size, shape, and encapsulation efficiency. Dynamic light scattering (DLS), a nanoparticle tracking analysis (NTA), VideoDrop interferometric imaging, and scanning electron microscopy (SEM) were employed for the determination of the size and shape of obtained nanoparticles. Zeta potential was measured by electrophoretic light scattering (ELS) to study differences in surface charge for NCs. The antimiR-25 in PG3 encapsulation efficiency (%) was quantified using a Quant-iT RiboGreen kit (ThermoFisher, Waltham, MA, USA).

### 3.5. Batch-Mode Malvern DLS Analysis

The hydrodynamic diameter (Dh), polydispersity index (PDI), and zeta potential (ZP) of NCs were measured by DLS and ELS (Zetasizer nano-ZS, Malvern Panalytical, Malvern, UK). The Zetasizer was equipped with a red 633 nm He–Ne laser and measurements were performed at a scattering angle of 173°. Measurements were performed either at 25 °C or a 25 °C–37 °C temperature ramp at increments of 0.5 °C/min. The refractive index (RI, 1.331) and viscosity (0.8872 cP) of water were used as dispersant properties. The laser power-attenuator was adjusted automatically. Size measurements were performed in disposable cuvettes, ZEN0040, and the determination of ZP in folded capillary zeta cells, DTS1070. Data was analyzed by Malvern Instruments Zetasizer Software version 7.13. Results are presented as the mean ± SD, *n* = 3.

### 3.6. Electron Microscopy Analysis

For imaging, NCs were placed on a copper grid and then either lyophilized (Christ alpha 2–4 ld plus, Osterode am Harz, Germany) or vacuum-dehydrated over silica gel overnight. Samples were sputter-coated with 20 nm gold (Leica EM SCD500, Wetzlar, Germany) and imaged by scanning electron microscopy (SEM, JSM-7001F, JEOL, Tokyo, Japan).

### 3.7. Nanoparticle Tracking Analysis (NTA)

The size and zeta potential of NCs were analyzed in Milli-Q^®^ water by a nanoparticle tracking analysis (NTA; ZetaView PMX120, Particle Metrix, Meerbusch, Germany). The laser wavelength was 520 nm. A ZetaView^®^ NTA—ZNTA cell was used. Measurements were performed over 1 cycle at 11 positions. ZP values were measured by pulses at −20 and +20 V. Measurements were performed in triplicates (n = 3) and results are shown as the mean ± SD.

### 3.8. Interferometric Imaging

NCs’ size was further assessed using an interferometric microscope (Videodrop, Myriade, France). A 7 μL sample was imaged with a maximum of 10 videos or a threshold of 300 particles. Measurements were performed in triplicates (n = 3) and results are shown as the mean ± SD. NCs’ sizes were determined with 0.28 mg/mL human holo-transferrin (HTF) (SIGMA, Burlington, MA, USA) in Milli-Q^®^ water.

### 3.9. Encapsulation Efficiency

NCs were prepared at pH 4, 5, and 6. To determine encapsulation efficiency (EE), 500 μL samples were centrifuged in a 10 kDa membrane Vivaspin tube (Sartorius, Goettingen, Germany) at 4000× *g* for 20 min. The filtrate was examined for free antimiR-25 using a fluorescence-based Quanti-IT kit (ThermoFisher, Waltham, MA, USA) in high-range measurement following the manufacturer’s instructions. The analyte was incubated with the Ribogreen fluorescent probe in a TE buffer and the fluorescence, excited at 480 nm, was read at 520 nm. Two calibration curves were made based on 2 different RNA types: ribosomal RNA (rRNA) provided with the kit and antimiR-25 corresponding to the investigated active pharmaceutical ingredient (API). The rRNA standard was diluted in the TE buffer to 1000 ng/mL, 500 ng/mL, 100 ng/mL, 20 ng/mL, and blank 0. For the antimiR-25 calibration curve, a stock solution of 286 ng/mL was diluted to 143 ng/mL, 71.5 ng/mL, 35.75 ng/mL, 23.83 ng/mL, and blank 0. Real blanks without Ribogreen reagents were prepared at 200 μL per well. Each standard was prepared in duplicates. Centrifuged samples (100 μL) were diluted with 100 μL of a reagent working solution. The plate was placed on a plate shaker for 5 min, protected from light. Given the 2× sample dilution, the measured concentration was doubled to obtain the final concentration. All data are normalized to 14% of the free antimiR-25, which passes through the Vivaspin membrane after centrifugation, and expressed as the mean ± standard deviation; measurements were performed in triplicates (n = 3).

### 3.10. Binding Constant (Ka) Between HTF and NCs

The aim was to determine the binding affinity between PG3 2/1 NCs and HTF, following the method of Chanphai et al. [27] based on protein UV absorption, taking advantage of the negligible dendrimer absorption at 280 nm. Briefly, a series of samples was prepared containing a constant concentration of protein HTF at 0.28 mg/mL while the concentration of NCs varied between 0.75 and 30 µg/mL. Absorbance at 280 nm of each sample was measured by UV spectroscopy (Nanodrop™ ND-1000, NanoDrop Technologies Inc., Wilmington, DE, USA); the respective binding constants were derived from the linear reciprocal plot of the absorbance as a function of dendrimer concentration as (Ka = intercept/slope). The Videodrop size analysis was performed as well.

### 3.11. In Vitro Studies: Bone Marrow-Derived Macrophages (BMDMs)

Macrophages were differentiated from non-manipulated wild-type C57BL/6 mice; bone marrow was isolated by flushing each femoral bone immediately following the bone cut. The cell suspension was centrifuged for 5 min at 350× *g*, resuspended, and passed through a cell strainer (70 µm). Cells were then counted and cultured in RPMI-1640 media supplemented with 10% Fetal bovine serum (FBS), Penicillin (100 U/mL), Streptomycin (100 µg/mL), HEPES (10 mM), 1X non-essential amino acid mix (ThermoFisher Waltham, MA, USA), 1× Sodium Pyruvate (Gibco, Waltham, MA, USA), β-mercaptoethanol (50 µM), and mouse recombinant M-CSF (Peprotech, Cranbury, NJ, USA) at a concentration of 10 ng/mL. Macrophages were differentiated in vitro in a supplemented RPMI 1640 medium as follows: for M0 and M2 macrophages, M-CSF was added at day 0, 3, and 5 at a final concentration of 10 ng/mL; for M2 macrophages, IL-4 (20 ng/mL) and IL-13 (20 ng/mL) (Immunotools Friesoythe, Germany) were added at day 5. After a further 2 days (day 7), macrophages were harvested, washed, and used as indicated.

### 3.12. Extracellular Vesicles

#### 3.12.1. GBM Cell Cultures

The mouse SB28 cell line was kindly provided by H. Okada, University of California, San Francisco (UCSF), USA. The cell line was cultured in serum-containing Dulbecco’s Modified Eagle Medium (DMEM) media (Gibco, Waltham, MA, USA) supplemented with 10% FBS that was centrifuged for 12 h at 100,000× *g* to deplete EVs. GBM cells were exposed to atmospheric O_2_ conditions in a conventional hood and incubator, or to 1% O_2_ using a Ruskinn 300 InVivO2 hypoxia workstation (Baker, Sanford, ME, USA) for 24–48 h. Media were pre-equilibrated to the desired oxygen level by flushing with the corresponding gas mix. SB28 cells tested negative for mycoplasma.

#### 3.12.2. EV Isolation and Characterization

EVs were isolated from supernatants of the SB28 mouse GBM cell line. Cells (2.5 × 10^5^) were cultured in 15 mL of a medium in T75 flasks (TPP) in triplicates for 24 h and the medium was collected for EV isolation. The isolation procedure was based on a previously described protocol with modifications [31]. Briefly, the culture medium was centrifuged at 300× *g* for 10 min to pellet the cells and large cell debris. The supernatant was then centrifuged for 10 min at 2000× *g* to remove dead cells and small cell debris. Finally, EVs were pelleted by ultracentrifugation at 100,000× *g* for 70 min and washed with PBS once, then pelleted again by ultracentrifugation at 100,000× *g* for another 70 min. The EV-containing pellet was resuspended in PBS for subsequent tests. The size and concentration of EVs were quantified using NTA (Particle Metrix, Inning am Ammersee, Germany). For further functional experiments, EV-to-recipient cell ratios were normalized to 3000 nanoparticles per recipient cell, as measured by NTA. The EV concentration used corresponds to 10^4^ EVs per macrophage, i.e., 6 × 10^8^ EVs to 6 × 10^4^ macrophages.

#### 3.12.3. Electron Microscopy and Cryo-EM

EV samples were placed on glow-discharged 200-mesh copper grids coated with formvar and carbon. After a 1 min absorption, samples were dried, washed three times, and stained with aqueous 2% uranyl acetate for 1 min. The stain was blotted dry from the grids with filter paper and samples were allowed to dry. Samples were then examined in a Tecnai 20 transmission electron microscope (FEI Company, Eindhoven The Netherlands) at an accelerating voltage of 80 kV. Digital images were obtained using the AMT Imaging System (Advanced Microscopy Techniques Corp., Woburn, MA, USA) (Appendix A).

Cryo-EM was used for the direct visualization of EVs. To prepare samples for the cryo-EM study, lacey carbon EM grids were glow-discharged (30 s, 25 mA) in a Pelco EasiGlow system. An aliquot (3 μL) of the EV suspension in PBS was applied to the carbon side of an EM grid, which was then blotted for 3.0 s and plunge-frozen into precooled liquid ethane. This procedure results in embedding the samples in a thin layer of amorphous ice to preserve them in their native state and to protect them from radiation damage. The samples were studied in a Talos Arctica (200 KeV, FEG) cryo-electron microscope (Appendix A).

### 3.13. SDS-PAGE and Western Blotting

SDS-PAGE was performed as follows. For cell lysates, protein concentrations were measured at 562 nm using a bicinchoninic acid assay kit (Pierce, Waltham, MA, USA), and 20 μg of protein was mixed with 4× NuPAGE LDS sample buffer. For EV extracts, proteins were also mixed with 4× NuPAGE LDS sample buffer. For reducing conditions, samples were supplemented with 10× NuPAGE reducing agent. All samples were then denatured at 95 °C for 10 min before being added to a 12% polyacrylamide Bis-Tris gel (Life Technologies, Carlsbad, CA, USA) and electrophoresed at 200 V for 70min in a MOPS SDS Running buffer (LifeTechnologies, Carlsbad, CA, USA). Following electrophoresis, the gel was transferred onto a polyvinylidene fluoride (PVDF) membrane using a Semi-Dry Blotting system (LifeTechnologies, Carlsbad, CA, USA) according to the manufacturer’s instructions. PVDF membranes were probed with primary antibodies specific for Hsp70 [EPR16892] (Abcam, Cambridge, UK), TSG101 [EPR7130(B)] (Abcam, Cambridge, UK), or Calnexin [EPR3633(2)]—an ER Membrane Marker (Abcam, Cambridge, UK) and Goat anti-rabbit secondary antibody conjugated with HRP (LifeTechnologies, 1:1000). The membrane was then incubated with an Amersham ECL Prime Western Blotting Detection Reagent for 5 min at room temperature and proteins were visualized using Fusion FX (Vilber Lourmat, Eberhardzell, Germany) (Appendix A).

### 3.14. BMDM Treatment with EVs and NCs

Cultures of BMDM, M0 and M2 phenotypes, were separately exposed to antimiR-25 and cGAMP NCs at different concentrations between 0 and 0.286 µg/mL for antimiR-25 NCs and 0 and 100 µg/mL for cGAMP NCs in 96-well plates (Corning, 3595, Somerville, MA, USA) for 24 h.

Independently, M0 and M2 BMDMs were first incubated in 96-well plates (Corning, 3595, Somerville, MA, USA) with a defined number of EVs for 24 h and then exposed to 0.143 µg/mL antimiR-25 and 100 µg/mL cGAMP NCs, separately and in combination for the next 24 h. Additionally, 0.9% saline and “empty” PG3 corresponding to 100 µg/mL cGAMP served as controls. Cells were supplied with 1%, 5%, or 21% oxygen levels throughout the whole incubation process in media that were preconditioned under each indicated O_2_ concentration for 24 h at 4 °C.

Lastly, M0 and M2 BMDMs were first incubated in 96-well plates (Corning, 3595, Somerville, MA, USA). cGAMP NC concentration was fixed at 50 µg/mL while antimiR-25 NC concentration varied between 0 and 1.43 ug/mL. PBS and “empty” PG3 corresponding to 50 µg/mL cGAMP served as controls. Cells were supplied with 1%, 5%, or 21% oxygen levels throughout all the incubation periods. After 24 h, the supernatants were collected and the levels of IFN-β were measured by ELISA.

### 3.15. ELISA-Based IFN-β Quantification

The corresponding supernatants were collected and assessed for Interferon β (IFN-β) content, and the experiments were executed as follows. Plates were coated with 50 µL/well anti-IFN-β RMMB-1 (PBL, 22400-1, Piscataway, NJ, USA) and incubated overnight at 4 °C. Next, standard dilutions of mouse IFN-β (PBL, 12400-1) and sample supernatants were added onto the coated plate (total: 50 µL/well). Then, the detection antibody anti-IFN-β was added (PBL, 32400-1), followed by secondary antibody anti-rabbit IgG, HRP-linked (Cell Signaling, 7074S, Danvers, MA, USA) addition. Importantly, each antibody addition was preceded by a washing step. Each incubation of the antibodies lasted at least 2 h. Finally, a colorimetric reaction was induced by adding 50 µL TMB substrate (BD OptEIA, 555214, Franklin Lakes, NJ, USA) and stopped by adding 25 µL of 1 M H_2_SO_4_. Optical absorbance was measured at 450 nm (including 570 nm—background noise cancelation) using a microplate reader (Biotek SynergyMx, BioTek Instruments, Winooski, VT, USA) to calculate IFN-β levels expressed as U/mL. Results are presented as the mean ± SD (n = 3).

### 3.16. RNA Extraction and qPCR

In total, 0.5 µg of total RNA isolated with a Total RNA Mini kit (A&A Biotechnology, Gdansk, Poland) was used to synthesize cDNA with a mix of random hexamers—oligo d(T) primers—and a PrimerScript reverse transcriptase enzyme kit (Takara bio Inc., Kusatsu, Japan) following the supplier’s instructions. SYBR green assays were designed using the program Primer Express v 2.0 (Applied Biosystems, Waltham, MA, USA) with default parameters. Amplicon sequences were aligned against the mouse/human genome by BLAST to ensure that they were specific for the gene being tested. Oligonucleotides were obtained from Invitrogen /Thermo Fisher (Waltham, MA, USA) (Appendix A). The efficiency of each design was tested with serial dilutions of cDNA. PCR reactions (10 µL volume) contained diluted cDNA, 2 × Power SYBR Green Master Mix (Applied Biosystems, Waltham, MA, USA ), and 300 nM of forward and reverse primers. PCR was performed on an SDS 7900 HT instrument (Applied Biosystems, Waltham, MA, USA) with the following parameters: 50 °C for two minutes, 95 °C for ten minutes, and 45 cycles of 95 °C for 15 s–60 °C for one minute. Each reaction was performed in three replicates on a 384-well plate. Raw Ct values obtained with SDS 2.2 (Applied Biosystems, Waltham, MA, USA) were imported in Microsoft Excel v2410 and the normalization factor and fold changes were calculated using the GeNorm method [32].

### 3.17. Statistical Analysis

The results were analyzed using a two-way analysis of variance (ANOVA) followed by Tukey’s or Sidak’s multiple comparison test. *p* < 0.05 was considered as statistically significant. * *p* < 0.05; ** *p* < 0.01; *** *p* < 0.001; and **** *p* < 0.0001.

## 4. Conclusions

Using various characterization techniques to analyze the designed nanoparticles, we selected nanocomplexes composed of antimiR-25 and glucoronidated PAMAM of the third generation with an N/P ratio of 2:1. This choice was supported by the optimal size of approximately 200 nm, which is conducive to macrophage uptake, and the negative zeta potential upon complexation, indicating the surface binding of antimiR-25 to PG3. We demonstrated the robustness of the nanocomplexes in critical formulation qualities, including resistance to pH variations within the range of 4 to 6, and tolerance to temperature elevations from 25 °C to 37 °C, confirming that the formulation meets the requirements for intravenous administration. Freeze drying was identified as the preferred storage technique for the nanocomplexes. The observed affinity between Tf and the antimiR-25 nanocomplexes suggests potential benefits for crossing the BBB and targeting GBM. Further investigation into cellular drug release mechanisms is needed, as the proton sponge effect may not be the primary contributor. This is suggested by the lack of pH impact on nanocomplex size and encapsulation efficiency.

In vitro biological assessments conducted on macrophages showed the efficacious induction of IFN-β secretion by cGAMP NCs even in conditions resembling hostile hypoxic zones of the TME, with high levels of GBM-derived EVs and low oxygen levels. Moreover, antimiR-25 and cGAMP NCs acted together to upregulate certain M1-associated genes. This highlights the potential of a multifocal approach in the modulation of the cGAS/STING pathway and the use of multicomponent NCs.

## Figures and Tables

**Figure 1 ijms-25-12787-f001:**
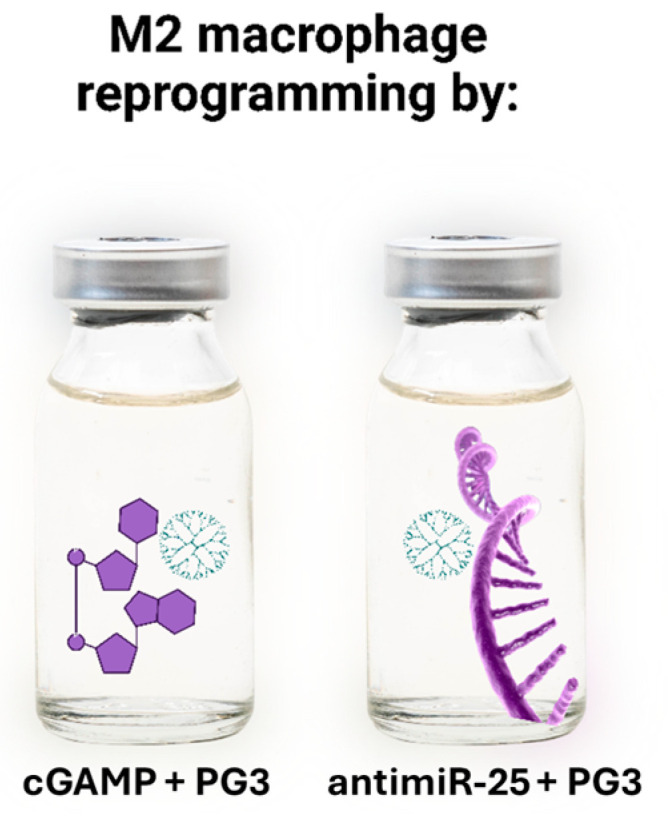
NCs made of cGAMP/PG3 and antimiR-25/PG3 NCS. Created in BioRender.

**Figure 2 ijms-25-12787-f002:**
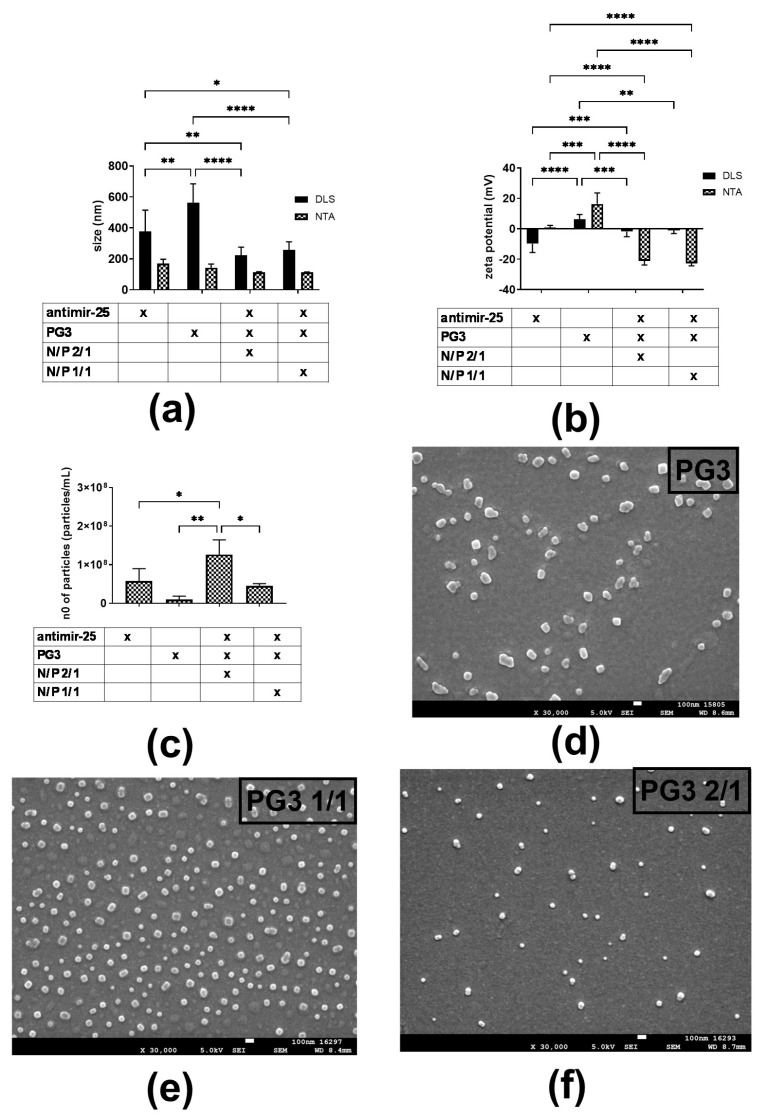
Nanocharacterization. (**a**) Zeta potential measured by ELS and NTA; (**b**) size measured by DLS and NTA, (**c**) number of particles measured by NTA; (**d**–**f**) scanning electron microscopy (SEM) images of antimiR-25 NCs. x-denotes the combination of formulation parameters being investigated Data are presented as mean ± SD of three independent experiments and comparisons were made using 2-way ANOVA. * *p* < 0.05, ** *p* < 0.005, *** *p* < 0.001, and **** *p* < 0.0001.

**Figure 3 ijms-25-12787-f003:**
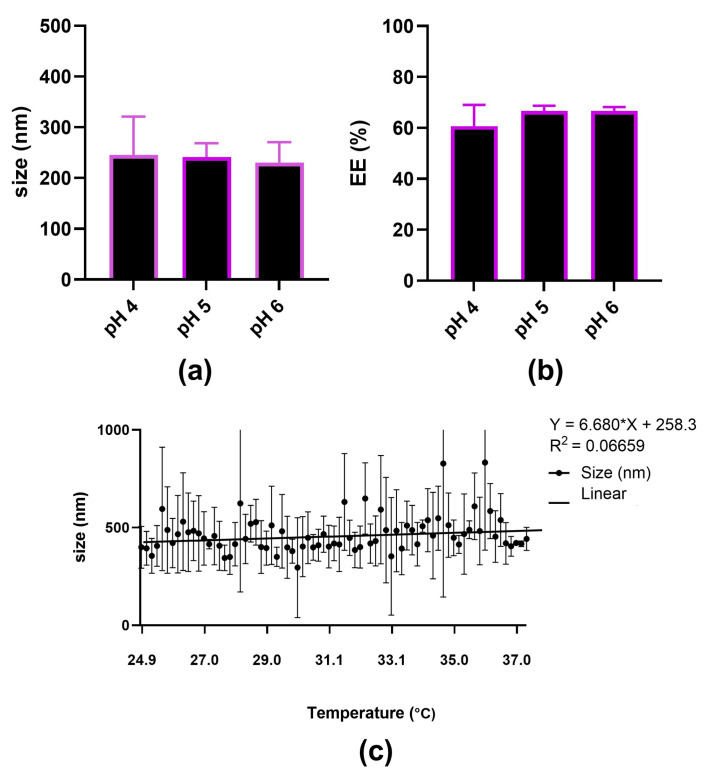
Critical formulation parameters. NCs were prepared in MilliQ water at pH 6. Additional pH values of 4 and 5 were adjusted before measurements. (**a**) Size measured at pH 4, 5, and 6 by DLS; (**b**) EE% measured at pH 4, 5, and 6 using a (Quant-iT RiboGreen assay kit (ThermoFisher, Waltham, MA, USA); (**c**) DLS size measurement of antimiR-25 NCs exposed to a temperature in the range of 25–37 °C.

**Figure 4 ijms-25-12787-f004:**
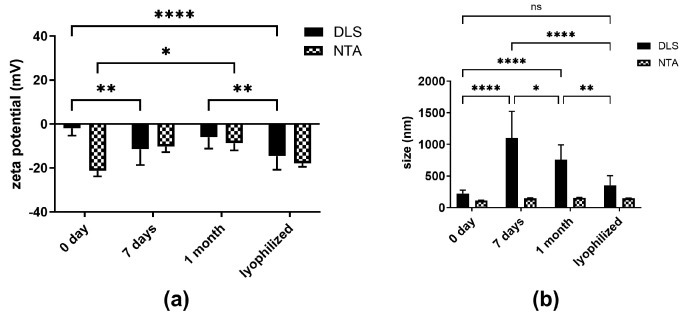
Storage stability (**a**) zeta potential and (**b**) size of antimiR-25 NCs measured by NTA and DLS at day 0 and 7 and 1 month or lyophilized. Data are presented as the mean ± SD of three independent experiments and comparisons were made using a 2-way ANOVA. * *p* < 0.05, ** *p* < 0.005, and **** *p* < 0.0001, ns means statistically non-significant.

**Figure 5 ijms-25-12787-f005:**
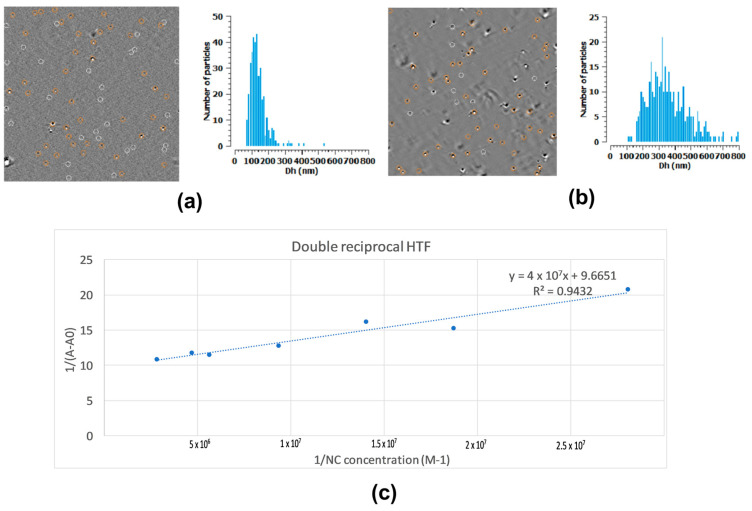
The antimiR-25 NCs’ interactions with transferrin. The Videodrop image capture and graphical representation of antimiR-25 NCs (**a**) and antimiR-25 NCs mixed with 0.28 mg/mL human holo-transferrin (HTF) (**b**); (**c**) binding affinity calculation between antimiR-25 NCs.

## Data Availability

Data is contained within the article and Appendix A.

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
