# Peer review of "Combining antimiR-25 and cGAMP Nanocomplexes Enhances Immune Responses via M2 Macrophage Reprogramming"

_ijms, 2024, doi:10.3390/ijms252312787_

Round 1

Reviewer 1 Report

Comments and Suggestions for Authors

This manuscript deals with the generation and characterization of antimiR-25/cGAMP nanocomplexes (NCs). The use of cGAMP NCs activated the cGAS-STING pathway in macrophages, leading to increased type I IFN (IFN-β) production and promoting a shift from M2 to 22 M1 phenotype.

In addition, the combo of cGAMP and antimiR-25 NCs further increase the expression of M1 markers. The authors conclude that these tools could be used for increasing the antitumor immune response in glioblastoma (GBM).

The authors performed a detailed analysis of the chemical and physical features of NCs charged complexed with the antimiR-25 and their functional effects on the production of IFN beta. This production is considered as the read-out assay for involvement of the cGAS-STING pathway.

The results clearly indicate that the cGMP NC can stimulate production of IFN beta (as already described by the same authors as referenced in previous reports. The effect of antimiR-25 on IFN-beta production are not evident. 

The combo culture condition (cGMP NC and antimiR-25NC)  (figure 9) can lead to a little increase (if any)  in IFN beta production, both at 1% or 21% O2 concentration.

The effects on M0 cells are evident, lesser evident on M2 cells. This does not support that the use of these NC (either alone or in combo) can indeed influence quite well the cGAS-STING pathway. This result is somehow contrasting the aim of this paper.

Overall, beside the authors show and performed a detailed analysis of NC, I would say that their functionality is really questionable. I understand the relevance of the generation of the new NC, but the biological relevance of the antimiR-25NC is far to be demonstrated. 

Some specific points.

1-figure 9. It is not clear whether the experiments have been performed at 1% and 21% of O2 concentration or at 5% O2 (instead of 1%) as indicated in materials and methods section (section 3.14. There is no mention of 1%O2, but just 5%O2). Please clarify better.

2. the analysis of IFN beta production has been tested after 24hrs of incubation at a given O2 concentration. It was not clear whether the BMDM cells have been cultured or maintained with a given O2 concentration before the triggering. I would imagine that a cell can sense and modify its behavior at a given O2 concentration if it stays in those conditions for some time to adapt it. Afterward, it can be tested the response of a cell to a stimulus. Please discuss. 

3. the M0, M1 or M2 polarization could be reinforced by the analysis of surface expression of macrophage markers. I perfectly understand that these markers are not exclusively (as well the analysis of gene expression) of a given population, but they can help to further characterize a macrophage subset.

4- the authors did not show any experiment aimed to demonstrate that NCs enter into BMDM. This is a really important point to understand well, the mechanism of action of at least cGMP-NCs.

5- I do not understand if in the preparation of the NCs it is not possible that cGMP and/or antimiR-25 can be free in the cultures during the cell culture. In other words, how much of the stimuli can be present in a soluble form during the cell culture?

6- the use of extracellular vesicles from the B28 GBM cell line is an interesting point. Please discuss the possibility that on these vesicles can be present, activating or inhibiting receptors or ligands for molecules present on BMDM.  Also, the possible presence of miR-25 in these vesicles is not shown.

7- the conclusion "Moreover, antimiR-25 and cGAMP NCs acted together to upregulate certain M1 associated genes."is not so suported by the data shown.

Author Response

This manuscript deals with the generation and characterization of antimiR-25/cGAMP nanocomplexes (NCs). The use of cGAMP NCs activated the cGAS-STING pathway in macrophages, leading to increased type I IFN (IFN-β) production and promoting a shift from M2 to 22 M1 phenotype.

In addition, the combo of cGAMP and antimiR-25 NCs further increase the expression of M1 markers. The authors conclude that these tools could be used for increasing the antitumor immune response in glioblastoma (GBM).

The authors performed a detailed analysis of the chemical and physical features of NCs charged complexed with the antimiR-25 and their functional effects on the production of IFN beta. This production is considered as the read-out assay for involvement of the cGAS-STING pathway.

The results clearly indicate that the cGMP NC can stimulate production of IFN beta (as already described by the same authors as referenced in previous reports. The effect of antimiR-25 on IFN-beta production are not evident.

Dear Madam or Sir, thank you very much for your comments!

As the reviewer pointed out cGAMP NCs alone could significantly stimulate IFN-β production. This lack of effect of antimir-25 can be attributed to the low endogenous levels of miR-25 in macrophages, meaning that any inhibition of miR-25 by antimir-25 is not readily detectable when IFN-β production is assessed. In contrast, when miR-25 is provided exogenously through GBM-derived EVs, the addition of antimir-25 has a significant impact on gene expression, such as Cxcl10 and Ifna (Figure 7), highlighting the context-dependent role of miR-25 in macrophage activation.

The combo culture condition (cGMP NC and antimiR-25NC)  (figure 9) can lead to a little increase (if any)  in IFN beta production, both at 1% or 21% O2 concentration.

The increase in INF-β production induced by the combination of cGAMP and antimiR-25 NCs is more pronounced in macrophage M0, which are more prone to be affected by exterior stimuli. They appear to be easier to shape in terms of immunological response regardless of the oxygen level, while macrophages M2 show lower immune response to the applied treatment. However, the trends are the same regardless of macrophage phenotype and applied conditions.

The effects on M0 cells are evident, lesser evident on M2 cells. This does not support that the use of these NC (either alone or in combo) can indeed influence quite well the cGAS-STING pathway. This result is somehow contrasting the aim of this paper.

Activation of the cGAS-STING pathway triggers an immune response characterized by the secretion of type I interferons, particularly IFN-β investigated here. By applying antimiR-25 and cGAMP NC, we clearly show efficacious induction of IFN-β by cGAMP NC (Figure 6) and we also observe functionality of the combination of antimiR-25 and cGAMP NC at the gene expression level. Based on expression of key genes, there was a shift in macrophage polarization from M2 to M1 and from M0 macrophages toward a proinflammatory M1 phenotype instead of the anti-inflammatory M2.

Overall, beside the authors show and performed a detailed analysis of NC, I would say that their functionality is really questionable. I understand the relevance of the generation of the new NC, but the biological relevance of the antimiR-25NC is far to be demonstrated. 

As detailed in our above replies, we claim functionality of both NCs particularly for expression of key genes associated with macrophage polarization, and high functionality of the cGAMP NCs for inducing IFN-β secretion.

Some specific points.

  1. figure 9. It is not clear whether the experiments have been performed at 1% and 21% of O2 concentration or at 5% O2 (instead of 1%) as indicated in materials and methods section (section 3.14. There is no mention of 1%O2, but just 5%O2). Please clarify better.

The experiments depicted in Figure 9 were performed under 1% and 21% O2. We have now updated the material and methods section to properly reflect all the conditions that are used throughout the manuscript) lines 548-564).

  1. The analysis of IFN beta production has been tested after 24hrs of incubation at a given O2 concentration. It was not clear whether the BMDM cells have been cultured or maintained with a given O2 concentration before the triggering. I would imagine that a cell can sense and modify its behavior at a given O2 concentration if it stays in those conditions for some time to adapt it. Afterward, it can be tested the response of a cell to a stimulus. Please discuss.

We agree with the reviewer that oxygenation levels can significantly impact macrophage polarization. However, it is difficult to choose an appropriate period of time over which to reduce oxygen levels, since this may destroy the baseline (M0) status that is used for comparison in the polarization studies. Indeed, when HIFs are stabilized under hypoxia, downstream effects are observed after a relatively short period, from 6 hours onwards. Therefore, to facilitate our analyses, we used seven days of cytokine stimulation, when macrophages had polarized toward M0 or M2. We then treated them with NC and/or EVs for 24 hours under 1%, 5%, or 21% O₂. Moreover, this culture used media preconditioned under each indicated O2 concentration, meaning that cells would immediately be exposed to the indicated oxygenation. These oxygen concentrations were selected to mimic hypoxic tumor zones (1% O₂), physioxic brain conditions (5% O₂), and standard laboratory conditions (21% O₂). This is now discussed in the manuscript (line 289-294).

  1. The M0, M1 or M2 polarization could be reinforced by the analysis of surface expression of macrophage markers. I perfectly understand that these markers are not exclusively (as well the analysis of gene expression) of a given population, but they can help to further characterize a macrophage subset.

We agree with the reviewer that surface marker expression can be an informative approach for assessing macrophage polarization status. In fact, in our previous publication (https://pubmed.ncbi.nlm.nih.gov/36145631/), we used flow cytometry and gene expression analysis to analyze M1- or M2-associated surface markers in BMDMs treated with NC. We were able to see a good correlation between surface marker expression (CD86 and CD206) and gene expression (Mrc1, Arg1, Stat1 and Nos2). In the present study, using the same protocol, we were able to determine changes in the gene expression in the macrophages treated with NCs

  1. The authors did not show any experiment aimed to demonstrate that NCs enter into BMDM. This is a really important point to understand well, the mechanism of action of at least cGMP-NCs.

We included a control using the empty nanocarrier (PG3), which showed no effect on INF-β levels. In contrast, the antimiR-25 and cGAMP-loaded nanocarriers stimulated INF-β secretion, indirectly indicating nanoparticle uptake. This suggests that the NCs indeed entered the BMDM. Our study primarily focused on the effects of the applied conditions rather than exploring the mechanism of action.

  1. I do not understand if in the preparation of the NCs it is not possible that cGMP and/or antimiR-25 can be free in the cultures during the cell culture. In other words, how much of the stimuli can be present in a soluble form during the cell culture?

Based on EE we can estimate how many antimiR-25/cGAMP are bound to NCs and the rest is in the soluble form. In our previous work, Figure 8, we have shown differences between soluble form of cGAMP and NCs (significantly different for M1 marker expression)

  1. the use of extracellular vesicles from the B28 GBM cell line is an interesting point. Please discuss the possibility that on these vesicles can be present, activating or inhibiting receptors or ligands for molecules present on BMDM.  Also, the possible presence of miR-25 in these vesicles is not shown.

We appreciate the reviewer’s interest in the role of EVs from GBM. The use of hypoxic SB28-derived EVs in our study was inspired by our previous findings (https://pubmed.ncbi.nlm.nih.gov/38389103/), which demonstrated that GBM cells upregulate miR-25 production under hypoxic conditions, secrete miR-25-loaded EVs, and transfer them to macrophages, potentially contributing to immunosuppression in the tumor microenvironment. We have routinely characterized and shown previously that hypoxic GBM cell derived EVs have high amounts of miR-25. Although our primary focus has been on hypoxia-induced EV cargo, specifically miR-25, we agree that other components in these EVs—such as surface proteins, lipids, and additional nucleic acids — may influence recipient cells through various mechanisms. We have now discussed these mechanisms in the manuscript (lines 46-68).

  1. the conclusion "Moreover, antimiR-25 and cGAMP NCs acted together to upregulate certain M1 associated genes."is not so suported by the data shown.

We understand and concur that the effects of the combined NCs are not dramatic under the conditions tested. However, as our detailed replies above argue, the value of the antimir-25 NC was not apparent in some of our assays. Nevertheless, we have established the feasibility of combining these NCs which do offer potential, after further refinement and testing, for future in vivo exploitation.

Reviewer 2 Report

Comments and Suggestions for Authors

Please, see the document below.

Comments on the Quality of English Language

The English could be improved to more clearly express the research.

Author Response

Authors Petrovic et al. provide an extensive and comprehensive study focusing on the use of antimiR-25/cGAMP nanocomplexes to modulate the immune response in GBM, with an emphasis on reprogramming macrophages from an M2 phenotype to an M1 phenotype. The work is scientifically sound and highly relevant to the field of cancer immunotherapy, especially given the aggressive nature of GBM with a focus on the TME.

Next, I attach a few suggestions that could improve the quality and readability of the manuscript:

  1. The main shortcoming of the manuscript is insufficient confirmation of hypotheses by in vivo experiments. For example, there is no direct confirmation of the passage of NCs through the BBB

Dear Madam or Sir, thank you very much for your comments!

We acknowledge the reviewer’s point regarding the need for in vivo validation, particularly with respect to the passage of nanocarriers (NCs) through the blood-brain barrier (BBB). In our current study, we propose a hypothesis about BBB passage, which we plan to investigate further in subsequent research. For the purposes of this manuscript, we focused on demonstrating the effects of cGAMP and anti-miR-25-loaded NCs on macrophage activation, particularly their ability to induce INF-β expression. In comparison, the empty PG3 nanocarrier showed negligible INF-β expression, highlighting the specific effects of the loaded cargo. While in vivo confirmation of BBB passage is an important next step, we believe that the current in vitro data effectively establish the relevance of the NC treatment and its potential therapeutic effects.

  1. The study addresses the proton sponge effect and cGAMP-mediated activation of STING but does not investigate alternative pathways or potential off-target effects of the nanocomplexes.

We attempted to address this sound question by tagging nanocomplexes. We considered attaching a dye to PG3, but doing so posed two risks: increased aggregation due to the saturation of terminal amino groups with mannose (glucuronic acid) and a potential reduction in available amino groups needed for binding cGAMP/antimir-25, which is essential for effective cell transfection.

  1. 3. The manuscript contains inconsistencies in the citation style is inconsistency, as seen in the references section. For example, the names of some journals are abbreviated, while others are given in full. This should be standardized according to journal guidelines.

Inconsistencies were corrected in the revised version of the manuscript – following uniformed ACS reference system.

  1. Please attach the raw membranes in triplicate to imaging methods such as TEM and WB and provide statistical evaluation/densitometric evaluation.

We have now included Supplementary figure 4 in the manuscript, which shows WB analysis and additional EM pictures.

Round 2

Reviewer 1 Report

Comments and Suggestions for Authors

The authors have replied to the reviewer's queries at least in part.

Reviewer 2 Report

Comments and Suggestions for Authors

Please, see the document below.

Comments on the Quality of English Language

The English could be improved to more clearly express the research.